# Interactions with Nature, Good for the Mind and Body: A Narrative Review

**DOI:** 10.3390/ijerph21030329

**Published:** 2024-03-12

**Authors:** Dahlia Stott, DeAndra Forde, Chetan Sharma, Jonathan M. Deutsch, Michael Bruneau, Jennifer A. Nasser, Mara Z. Vitolins, Brandy-Joe Milliron

**Affiliations:** 1Nutrition Sciences Department, College of Nursing and Health Professions, Drexel University, Philadelphia, PA 19104, USA; dps85@drexel.edu (D.S.);; 2Department of Kinesiology, Health, Food & Nutritional Sciences, University of Wisconsin-Stout, Menomonie, WI 54751, USA; 3Department of Food and Hospitality Management, College of Nursing and Health Professions, Drexel University, Philadelphia, PA 19104, USA; 4Health Sciences Department, College of Nursing and Health Professions, Drexel University, Philadelphia, PA 19104, USA; 5Wake Forest University School of Medicine, Wake Forest University, Winston-Salem, NC 27101, USA

**Keywords:** interaction with nature, green space, mental health, physical health, health behaviors

## Abstract

Interacting with nature may promote mental and physical health. There are multiple ways to interact with nature: indirectly, incidentally, and intentionally. How these types of interactions with nature may be associated with mental and physical health status and health behaviors is unclear. The purpose of this narrative review is to (1) describe the relationship between interactions with nature (indirect, incidental, and intentional) and mental and physical health outcomes and behaviors, (2) identify gaps in the literature, and (3) provide recommendations for future research. Considerable evidence suggests that interacting with nature, indirectly and intentionally, is associated with improvements in mental health and physical exhibitions of mental status. Furthermore, intentionally interacting with nature is associated with engagement in physical activity and gardening is associated with fruit and vegetable consumption. Research suggests that incidentally interacting with nature may be associated with positive mental health status. More research is needed to understand the relationships between incidental interactions with nature and physical health status and behaviors; as well as among all types of interactions with nature and physical health disorders, sleep, and dietary behaviors.

## 1. Introduction

Interacting with nature, or the natural world including plants and landscapes, is being recognized as an important public health intervention to help manage chronic health conditions. Numerous programs have been developed to encourage healthcare providers to prescribe time in nature to their patients to aid in the treatment of mental health conditions such as anxiety and depression and physical diseases such as obesity and type 2 diabetes mellitus. One such program, Park Rx America, provides clinicians and patients with a digital tool for patients to be reminded to “fill” their prescriptions and to log the time they spend in nature [1]. Similarly, some doctors in Canada are providing their patients with free passes to national parks when prescribing time in nature [2]. Throughout the world, there are additional practices that encourage people to spend time in nature. For example, Shinrin-yoku, also called “forest bathing”, is a practice where people intentionally engage their senses while being immersed in a forest [3]. While the term Shinrin-yoku was first coined in Japan in the 1980s, this practice has been implemented in other countries such as Singapore, Switzerland, Germany, and the United Kingdom [4]. These programs and practices are informed by research which indicates that interacting with nature promotes better mood and decreases depression, anxiety, stress, and blood pressure levels [5,6,7].

Being able to simply observe nature has also been associated with improved health outcomes. Dr. Roger Ulrich, a leader in evidence-based healthcare design, is best known for his research in how integrating nature in the clinical setting improves patient outcomes. In one such study, patients with a view of nature had a shorter hospital stay, took fewer doses of moderate and strong pain-relieving medications, and had fewer postoperative complications, when compared to those who had a view of a brick wall [8]. The effects of nature on patients’ recovery have continued to be explored and this has led to nature (e.g., plants and pictures of nature) being incorporated within clinical settings [9].

Health behaviors may also be influenced by interacting with nature [7]. Physical activity, sleep hygiene, and a nutritious diet are examples of healthy behaviors, whereas smoking and being sedentary are examples of unhealthy behaviors. The joint effect of engaging in healthy and unhealthy behaviors contributes to mental and physical health outcomes such as depression, anxiety, and stress; and self-reported health, cardiovascular disease, and high blood pressure [10,11]. Thus, focusing on health outcomes is not enough; practicing healthy behaviors will further promote beneficial health outcomes [10].

There are many ways to interact with nature. Taking these into account, Keniger et al. categorized three types of interactions with nature: indirect, incidental, and intentional interactions (Table 1) [12]. Indirect interaction with nature is defined as experiencing nature without being in nature and includes viewing nature through a window and looking at pictures of nature. Incidental interaction with nature is experiencing nature secondary to the main activity being performed and includes walking through or by nature and taking care of houseplants. Intentional interaction with nature is defined as purposeful interaction with nature and includes visiting a park, exercising outdoors, and gardening.

Previous reviews have primarily focused on one type of interaction with nature, predominantly exploring the benefits of intentional interactions with nature for health and wellbeing [3,5,6,7,26]. While there are existing reviews on the health benefits of exposure to nature, to the best of our knowledge, this is the only review that examines all three types of interactions with nature and their associations with mental and physical outcomes as well as health behaviors, in the last 10 years. Therefore, the purpose of this narrative review is to (1) describe the relationship between indirect, incidental, and intentional interactions with nature and mental and physical health outcomes and behaviors, (2) identify gaps in the literature, and (3) provide recommendations for future research.

## 2. Materials and Methods

A narrative review was conducted following guidelines set forth by Green et al. [27]. Articles relating to interactions with nature, mental health, physical health, and health behaviors were selected for this narrative review. The search for published literature from PubMed, PsycINFO and ancestry-search methods (e.g., references in articles) occurred from October 2022–November 2023. The Boolean for PubMed was ((greenspace OR green space) OR (nature-based) OR (urban green space) OR (urban greenspace) OR (outdoors) OR (houseplant) OR (garden)) AND (((“Mental Health”[Mesh] OR mental health) AND (“Depression”[Mesh] OR depression) OR (“Anxiety”[Mesh] or anxiety) OR “Depressive Disorder”[Mesh] OR (“Stress Disorders, Traumatic, Acute”[Mesh] OR Stress) OR (“Heart Rate”[Mesh] OR heart rate) OR (“Physiology”[Mesh] OR physiology) OR (“Blood Pressure”[Mesh] OR blood pressure) OR (“Health”[Mesh] OR health) OR (“Quality of Life”[Mesh] or quality of life)) AND ((“Sleep”[Mesh] OR sleep) OR (“Exercise”[Mesh] OR exercise OR physical activity) OR (“Diet, Food, and Nutrition”[Mesh] OR nutrition OR diet))). The Boolean utilized in PsycINFO was (TI [title] greenspace OR AB [abstract] greenspace OR TI green space OR AB green space OR TI nature-based OR AB nature-based OR TI outdoor* OR AB outdoor* OR TI houseplant* OR AB houseplant* OR TI garden* OR AB garden*) AND ((“Mental health” OR “Depression” OR “Anxiety” OR “Stress” OR “Heart rate” OR “Physiology” OR “Blood Pressure” OR “Wellness” OR “Quality of Life”) OR (health behavior* OR “physical activity” OR Exercise* OR Nutrition* OR Diet*)). Limiters were used for PubMed and PsycINFO. The limiters for PubMed included articles published 2013–2023, humans, adults 19+, and articles in English. The limiters for PsycINFO were as follows: Published Date: 2013–2023; Peer Reviewed; Publication Type: Peer Reviewed Journal; English; Age Groups: Adulthood (18 years and older); Exclude Dissertations.

Only peer-reviewed articles from the previous 10 years (2013–2023) that included quantitative original research, an adult population, and were published in English were included in this review. Additional inclusion criteria were articles that measured individuals’ perceptions of the nature around them or reported the duration of time spent in or with nature, as opposed to measuring the actual amount of greenspace surrounding participants’ homes (e.g., Normalized Difference Vegetation Index and Geographic Information System). For example, participants in cross-sectional studies may have self-reported the nature outside of their window and participants in intervention studies may have engaged in nature-based therapy. Papers that were excluded were descriptive or qualitative papers, articles that measured outdoors without explicit mention of nature (except within the context of physical activity, as per the work by Keniger et al.), and papers where the authors did not compare outdoor physical activity to indoor physical activity.

The lead author (D.S.) read the titles and abstracts of the papers and provided judgment on the inclusion of each article. These were then verified by another author (D.F., C.S., or B.J.M.). Disagreements were mediated and decided by B.J.M. Data were manually extracted by two authors (D.S. and D.F.) into a sheet including the lead author, year of publication, population, sample size, location, study type, health outcome or behavior measured, intervention description (if applicable), relevant results, and type of interaction with nature.

## 3. Results

The following section summarizes the literature in the last ten years that explores the relationships between interactions with nature (indirect, incidental, intentional) and mental health, physical health, and health behaviors. Within each subsection, we present observational and then experimental results.

### 3.1. Overview of Identified Studies

The online search retrieved 3841 articles for review. From PubMed, 2477 articles were yielded, PsycINFO yielded 1304 papers, and 60 papers were identified through ancestry methods. After reviewing titles and abstracts, 266 were identified as duplicates and 541 were identified as meeting inclusion criteria. Sixteen articles were unable to be retrieved. A further 249 papers were excluded after review of the full-length article due to these papers not meeting inclusion criteria. A total of 276 papers were included in this review. The process of screening and selection of articles for this review paper is shown in Figure 1.

Fifty-four studies were conducted in Asia, with China (n = 19) and Japan (n = 12) representing the countries from this region that most often studied the relationships between interactions with nature and health. One hundred and six studies were conducted in European countries, including the United Kingdom (n = 30) and Germany (n = 11). Seventy-five studies were conducted in North America: 64 from the United States and 11 from Canada. Seventeen studies were conducted in Oceania: 15 from Australia and two from Aotearoa. Five studies were conducted in South America, four of these being from Brazil. Seven studies were conducted in the Middle East: Iran (n = 4), Israel (n = 1), Pakistan (n = 1), and Saudi Arabia (n = 1). Three studies were conducted in Africa: Morocco, Nigeria, and Ethiopia. Nine studies collected data in multiple countries and regions of the world.

A total of 53 papers studied indirect interactions with nature, 11 studied incidental interactions, and 202 studied intentional interactions. Two papers studied indirect and incidental interactions, six papers studied indirect and intentional interactions, one paper studied incidental and intentional interactions, and one paper studied all types of interactions with nature. While all articles studied adults, some studied specific populations including cancer survivors, incarcerated individuals, and students. Cross-sectional (n = 116), longitudinal (n = 29), intervention (n = 126), and mixed method (n = 5) studies were identified in this review. The Appendix A displays the type of interaction with nature studied, the studies included in this narrative review, the specific population studied, the location of the study, type of study, the intervention (when applicable), and the relevant results. Figure 2 displays the types of interactions with nature studied across the world.

Studies included in this review controlled for potential confounders including age, race, ethnicity, gender, income, connection to nature, and body mass index (BMI). We were not able to identify articles that studied incidental interactions with nature and health behaviors.

### 3.2. Indirect Interactions with Nature

#### 3.2.1. Mental Health

Most research that has explored indirect interactions with nature has studied views of nature from indoors. This type of interaction at work and home seems to be beneficial to cognition and mental health. Within workplaces, being satisfied with a natural view through a window has been associated with mental wellbeing [28], 2.13 greater odds of work ability, and 3.03 greater odds of job satisfaction [29]. Likewise, having a window with a view of nature has been associated with greater concentration [30], less job stress [31], and decreased health complaints [31]. For healthcare professionals working during the COVID-19 pandemic, having a view of nature was inversely associated with emotional exhaustion, depersonalization, reduced personal accomplishment, and burnout [32].

Having a view of nature at home or in living spaces may be beneficial to mental health. Multiple studies have reported that viewing nature from the living space is positively associated with distress tolerance [13], life satisfaction [13,33,34], self-esteem [33], happiness [33], and wellbeing [13]; and inversely associated with depression [13,33,34], anxiety [13,33,34], loneliness [13,33], and negative affect (negative emotions) [35]. In contrast, having a limited view or no view of nature at home has been associated with experiencing depression [36,37] and poorer wellbeing [38]. The perceived amount of greenspace has also been significantly associated with better mental health [39,40]; greater emotional wellbeing [40,41]; greater quality of life [42]; lower perceived stress, depression, and anxiety [43]; fewer symptoms of psychological distress [44,45]; and decreased odds of serious mental illness [44]. Likewise, perception of greenspace has been reported as a significant predictor of psychological wellbeing [46]. Additionally, satisfaction with greenspace has been associated with positive mental health [47] and quality of life [48]. Many of the studies which presented these findings were undertaken during the COVID-19 pandemic [13,33,34,37,43], a time of greater distress and when stay-at-home measures may have been implemented. Therefore, these findings should be confirmed when the population is not in a crisis situation.

Researchers have aimed to improve mental health through indirect interactions with nature. Their interventions have utilized pictures of nature [14,49,50,51,52], virtual reality [15,53,54,55,56], videos [16,49,57,58,59,60], nature sounds [17,61,62], and guided imagery [18]. After indirectly interacting with nature, participants of these interventions experienced greater positive affect (positive emotions) [14,15,16,53,55,56,59], happiness [49], satisfaction with life [59], emotional response [51], mood [52], and perceived restorativeness [16,56]; and decreased negative affect [15,53,58,60], anxiety [17,18,57,61,62], depression [57,63], rumination [58], and agitation [17,61]. Chang et al. utilized functional magnetic resonance imaging in their study and reported that viewing pictures of urban green landscapes activated regions of the brain that have been associated with executive attention [50]. Though there are promising results that indicate that indirectly interacting with nature promotes mental health, results from some intervention studies do not support this premise. Anderson et al. reported that, for individuals who were on isolated deployment, viewing nature through virtual reality did not result in any significant changes in positive or negative affect [54]. Participants who watched a video of nature during their work break did not experience any significant difference in directed attention or problem-solving when compared to those that took an unstructured break [64].

#### 3.2.2. Physical Health

Only two studies reported a positive relationship between indirectly interacting with nature by having a view of nature and better self-reported health [21,47]. Multiple researchers have examined the impact of viewing nature on physical health as stress not only exhibits itself psychologically but also physically (e.g., sympathetic nervous activity dominance, greater heart rate and blood pressure, and increased alpha amylase and cortisol). In experimental studies, researchers have measured changes in physiology before and after indirectly exposing participants to nature through views of nature [65,66], virtual reality [15,53,67,68], videos [57,69], and sounds [17,68]. In several studies, after being indirectly exposed to nature, participants experienced greater parasympathetic nervous system dominance [65,66,68], heart rate variability [65,69], and decreased heart rate [15,53,66,68,69], systolic and diastolic blood pressure [17,53,67], skin conductance [53,65], pain [57], and cortisol [53]. In one study, after five minutes of viewing nature using virtual reality, the alpha amylase concentration decreased by 1.2 ng/mL and diastolic blood pressure by 4.6 mmHg [53]. In another study, salivary alpha amylase significantly increased after viewing nature via virtual reality but this was observed in one of the seven nature conditions and was not observed in the other conditions [67]. In addition, diastolic blood pressure decreased in the same nature condition, indicating that the participants may not have been experiencing stress [67]. The duration of indirect exposures to nature in these interventions ranged from 5 to 45 min, indicating that indirectly interacting with nature, even for a short amount of time, has measurable benefits to physical health.

#### 3.2.3. Health Behaviors

Only 17 studies examined the relationships between effects of indirect interactions with nature and health behaviors. Cross-sectional studies have reported a positive relationship among the perceived amount of greenspace, walking [70,71], frequency and duration of physical activity [72], and hours engaged in moderate to vigorous physical activity [19]. However, Ali et al. reported that there was no significant relationship between perceived amount of greenspace and engagement in physical activity [73]. It has also been reported that perceived distance to a greenspace has been associated with walking [74] and difficulty getting to a greenspace has been associated with lower engagement in physical activity [75].

Exercising while indirectly interacting with nature may further promote mental health compared to exercising indoors. Participants of multiple interventions were indirectly exposed to nature while exercising through viewing pictures [76] or videos [77,78] of nature. In these studies, participants experienced greater pleasure [77], decreased mood disturbance [76], greater parasympathetic nervous system activity [79], and reported that the experience was calm and positive [78]. Compared to those who prefer exercising indoors, those who prefer outdoor exercise have exhibited greater attention and lower stress after looking at pictures of nature [80]. In comparison to the above findings, Ahnesjö et al. reported that, in their experiment, there was no significant difference in heart rate between participants who exercised in a simulated outdoor environment and an indoor environment [81]. This result may be explained because the participants engaged in low-intensity exercise in both conditions [81].

Additional research has also suggested that simply having a view of nature may influence other health behaviors. For example, results from a cross-sectional study indicated that having a view of nature through a window was associated with a decreased risk of poor sleep quality [30]. In another study, having a view of nature was associated with decreased frequency of food cravings and the strength of cravings, and these relationships were mediated by negative affect [35]. Michels et al. had groups of participants view pictures of plants or non-nature objects over a six-minute period and measured participants’ desire to consume fruits, vegetables, and snacks [14]. The results from this study indicated that viewing plants was significantly associated with an increase in desire to consume vegetables and a decrease in wanting to consume and preference for snacks; there was no significant change in participants’ desire to consume fruits [14]. Catissi et al. reported that, after watching a video of nature, participants experienced improvements in tiredness and appetite [57]. Though these results signify that indirectly interacting with nature may promote healthy behaviors, more research is needed to confirm these results.

### 3.3. Incidental Interactions with Nature

#### 3.3.1. Mental Health

The type of incidental interaction with nature most studied is the presence or care of houseplants. Results from a study conducted in China during the COVID-19 pandemic indicated that having houseplants may be positively associated with fewer symptoms of depression and anxiety [20]. Maury-Mora et al. also conducted a study during the COVID-19 pandemic and reported that, when compared to individuals who had access to greenspaces at home, those that had indoor plants experienced fewer symptoms of stress [82]. Furthermore, mental wellbeing and mindfulness have been positively associated with hours and years of caring for houseplants, and the number of houseplants in the home [83]. Likewise, having houseplants has been associated with emotional wellbeing [38,84]. The potential benefit of caring for houseplants to wellbeing has also been reported in experimental studies. After caring for indoor plants, participants reported greater feelings of relaxation and comfort [85]; improved happiness, calmness, and peacefulness [86]; and decreases in mood, stress, depression, and trait anxiety [87]. Having indoor nature may also benefit employees as satisfaction with the indoor environment in a green building has been associated with greater wellbeing [88] and indoor contact with nature at work has been associated with decreased job stress and subjective health complaints [31]. Heilmayr and Friedman conducted an intervention over four weeks to promote wellbeing which utilized five intervention groups: moderate indoor exercising, social film club, exposure to nature, community gardening, and taking care of indoor plants. While wellbeing improved among all participants, there was no significant differences between groups [89]. Limited evidence indicates that commuting through a natural environment is associated with better mental health [90].

#### 3.3.2. Physical Health

We were only able to identify four studies that examined how incidentally interacting with nature may be associated with physical health. In a study among elderly adults in Taiwan, houses that did not have houseplants had significantly more particulate matter and total volatile compounds in the air than homes that had houseplants [91]. Moreover, a greater amount of particulate matter and total volatile compounds was further associated with greater heart rate and blood pressure [91]. These results indicate that the presence of houseplants may promote cardiovascular health for older adults. Lee et al. measured sympathetic nervous system activity and diastolic blood pressure in their experimental study [85]. They reported that, after transplanting a houseplant, participants had significantly less sympathetic nervous system activity and diastolic blood pressure compared to when they completed a computer activity [85]. In Pedrinolla et al.’s intervention, participants walked in an indoor nature environment for multiple weeks and experienced a significant decrease in salivary cortisol [92]. Cox et al. reported that working outdoors was associated with greater self-reported health [21].

### 3.4. Intentional Interactions with Nature

#### 3.4.1. Mental Health

Many researchers have studied the relationship between intentional interactions with nature and mental health. Results from their research suggest that intentionally interacting with nature is positively associated with mental wellbeing [93,94,95,96], mood [97,98], calmness and wakefulness [99], subjective wellbeing [100,101,102,103], positive affect [95,96,104,105], quality of life [106,107,108], vigor [98], mindfulness [105], satisfaction with life [109,110], vitality [111], happiness [112,113], relaxation [114], and restorativeness [115]. Intentionally interacting with nature has been inversely associated with depressive symptoms [116], depression [95,96,98,106,110,117], anxiety [97,110], stress [94,95,96,105,118], negative affect [94,95,96,104,105], anger [97], agitation [106], fatigue [98], and risk of cognitive impairment [119,120]. The frequency of intentionally interacting with nature has been found to be positively associated with mental wellbeing [22], emotional health [117], mental health [117,121], and inversely associated with depression [96,122], mental distress [22,123,124], stress [96,123], negative affect [96], and the use of medication to treat depression [22]. The duration of intentional interactions with nature has also been positively associated with positive affect [96], mental wellbeing [125,126,127], subjective wellbeing [128], mental health [111,129], vitality [130], life satisfaction [126,128], quality of life [131], and inversely associated with depression [122,125,132], with Haider et al. identifying these relationships when individuals spend ≥ 60 min outdoors [125].

According to recent analyses, it has been suggested that intentionally interacting with nature for at least 30 min per week could prevent 7% of depression cases [133] and that spending at least two hours a week in nature may be associated with wellbeing [134]. Making an effort to experience nature (e.g., taking pictures, smelling, touching) has also been associated with positive affect [135]. Marselle et al. reported that participating in nature group walks reduced participants’ depression from recent stressful events [95]. Decreasing the amount of time intentionally interacting with nature may be deleterious for mental health as when compared to individuals who had no change in time intentionally interacting with nature, those who reported less time intentionally interacting with nature reported higher depression [136], anxiety [136], and stress [123] during the COVID-19 pandemic.

Several authors have reported results contrary to the general literature including that interacting with nature was not associated with wellbeing [137], depressive symptoms [138], quality of life [139], and cognition [140]; and was associated with worse mental health [141,142], greater depression [143], and greater stress [115,144]. It has also been reported that there are no differences in mental health status between gardeners and non-gardeners [145]. Young et al. reported that increasing time outdoors was associated with greater anxiety; however, it is important to note that this study occurred during the COVID-19 pandemic [136]. In Olszewska-Guizzo et al.’s longitudinal study over the COVID-19 pandemic, participants who intentionally interacted with nature more had a significant decrease in frontal alpha asymmetry, a brain wave indicative of positive emotions [146].

The evidence suggesting a positive relationship between intentional interactions with nature and mental health has led to the development of experimental studies. Improvements in mental health have consistently been reported, as indicated by increases in wellbeing [147,148,149,150,151,152,153,154,155,156,157] and positive affect [55,56,147,148,149,150,158,159,160,161,162]; and decreases in depression [149,151,152,163,164,165,166,167,168,169], anxiety [148,149,151,154,159,163,165,166,169,170,171,172,173,174], stress [148,158,159,165,169,175,176,177,178,179,180,181], and negative affect [147,148,150,159,182,183]. Many of these same intervention studies have resulted in improvements in other aspects of mental health such as relaxation [172,173,179,184,185,186,187], fatigue [163,166,170,171,172], quality of life [164,176,188,189], vigor [162,163,170,172,190], restorativeness [56,162,176,191], anger [163,170,171,172], mood [52,180], total mood disturbance [174,192], self-compassion [166,193], confusion [163,170,171,172], cognition [194,195] and distress [180,196], in addition to numerous other aspects and measurements of mental health (see Appendix A) [147,148,149,151,152,154,161,166,167,177,179,187,188,197,198,199,200,201,202,203]. Interventions that incorporate intentional interactions with nature have also demonstrated an improvement in post-traumatic stress disorder symptoms [165] and postpartum depression [174]. In another recent intervention study, individuals who received modified mindful-based cognitive therapy in nature over eight weeks had decreased depression relapses and fewer weeks with major depressive disorder when compared to those who received usual care [164]. Only a few intervention studies were identified with contradictory results such as no change in quality of life [204] and no difference in stress [205] and mental health outcomes [206] between intervention and control groups.

#### 3.4.2. Physical Health

Research suggests that intentionally interacting with nature is positively associated with physical health such as greater self-rated health [21,98,117,207,208,209,210], wellness [207], physical function [211], resolved knee pain [212], sperm concentration [213], and mobility [214]; and lower blood pressure [129], diastolic blood pressure [215,216], BMI [98,217,218], weight [98], waist circumference [216], triglycerides [216], cortisol [219], cardiovascular disease [117,129], and cardiometabolic risk [216]. Intentionally interacting with nature has also been associated with decreased odds of cardiovascular disease, stroke, heart attack, high cholesterol, high blood pressure, diabetes, BMI ≥ 25, poor physical health status, high 10-year mortality risk [121]; and decreased risk of developing frailty [220], lower limb and hip fractures [221], and mortality [222,223,224]. Using dose response modeling, Shanahan et al. reported that 9% of high blood pressure cases could be averted if individuals spent at least 30 min outdoors each week [133]. Furthermore, spending two hours a week intentionally interacting with nature has been associated with good health [134]. Some researchers reported that intentionally interacting with nature was associated with higher BMI [117], was not associated with arterial stiffness [225], and that there was no difference in physical health between gardeners and non-gardeners [145].

Similar to indirect interactions with nature, the effect that intentional interactions may have on physiological exhibitions of stress has been studied. Cortisol is a common biochemical marker used to assess levels of stress, especially because cortisol can be stable over a three-week period [158]. Researchers of several experimental studies have measured changes in cortisol before and after spending time in nature. Several studies have reported significant decreases in cortisol after participants spent time in nature [158,177,226,227,228]. Furthermore, Hunter et al. was able to estimate how much time in nature is needed to decrease cortisol; reporting that intentionally interacting with nature for 20–30 min resulted in an 18.5% decrease in cortisol per hour [229]. Cortisol continued to drop after spending more than 30 min intentionally interacting with nature but at a decreased rate [229]. In a six-month intervention study, cortisol decreased on intervention days but not over the entire intervention period [230]. In contrast, Biel and Hanes were unable to detect any differences in cortisol concentration after participants sat in one of four environments ranging from very natural to very built, likely due to the small sample size (n = 15) [231].

Additional intervention research studies have reported improvements in parasympathetic nervous system activity [170,171,172,173,232], heart rate variability [233], heart rate recovery [234], and general health [23,235]; and reductions in sympathetic nervous system activity [170,171,172,236], heart rate [170,171,172,184,232,237,238,239], overall blood pressure [184], systolic blood pressure [162,171,179,227,234,237,238], diastolic blood pressure [162,179,227,234], cardiac function [237], vascular function [237], mean arterial pressure [162], pulse pressure [237], C-reactive protein [226], interleukin 6 [226], gastrointestinal symptoms [240], and pain [235]. Researchers have also shown that, after interventions that utilize intentional interactions with nature, there are significant increases in brain-derived neurotrophic factor [195,241] and platelet-derived growth factor [241], which are related to memory. Sudimac et al. conducted an experimental study that measured the activity of the amygdala, which is greater during times of stress, and reported that participants experienced less amygdala activity when faced with a stressful situation after walking in a natural environment, compared to those who walked in an urban environment [242]. In one study, participants who sat in a very built setting experienced an increase in salivary alpha amylase when compared to those who sat in mostly built, mostly natural, and very natural environments [231], showing that being in a built and non-nature environment increases alpha amylase. In another experiment, salivary alpha amylase concentrations increased and there were no changes in heart rate variability observed over the six month intervention period, which indicated that the intervention may have increased participants’ physiological exhibitions of stress [230]. The variation in alpha amylase concentrations between the participants and the small sample size (n= 11) could explain these results.

#### 3.4.3. Health Behaviors

Intentionally interacting with nature is not only associated with health outcomes but also health behaviors, especially engagement in physical activity [49,113,207,219,243,244,245,246]. For example, greater frequency of intentional interactions has been associated with meeting physical activity guidelines [190,247,248], more steps taken per day [249], and engagement in moderate-to-vigorous physical activity [250]. Greater durations of intentional interactions with nature have been associated with engagement in physical activity [133] and moderate-to-vigorous physical activity [251]. It has also been reported that park visitors have greater engagement in moderate-to-vigorous physical activity than non-park visitors [252]. Razani et al. conducted an intervention where participants received a park prescription and, at the end of the experiment, the participants significantly increased their moderate-to-vigorous physical activity [177]. Likewise, researchers have demonstrated that outdoor exercise interventions can lead to significantly greater increases in physical activity when compared to indoor exercise [168,253].

Cross-sectional research suggests that exercising in nature enhances the benefits of physical activity. Individuals who exercise outdoors have greater wellbeing [254,255,256,257], general health [255,258], positive affect [257], happiness [259], stress management [258], and quality of life [259]; and lower depression [260,261,262], anxiety [261,262,263], somatic anxiety (the physical manifestations of anxiety, such as abdominal pain, rapid or irregular heartbeat, fatigue, or insomnia) [256], stress [259], heart rate [264], and triglycerides [264]. Greater duration of engagement in outdoor physical activity has been associated with greater quality of life and subjective happiness but not anxiety among cancer survivors [24]. Van den Heuvel et al. also reported that duration of outdoor exercise was not associated with vitamin D status (which is synthesized when skin is exposed to UVB rays [265]); though van den Heuvel et al. explained that the intensity of physical activity may have a greater effect on vitamin D status rather than the location of the exercise [266]. Researchers who report interventions that compare the health effects of outdoor physical activity to indoor physical activity have described significant increases in positive affect [267,268], and psychological quality of life [269]; and significant decreases in diastolic blood pressure [267], depression [168], and cortisol [267]. Some researchers have reported no significant difference between exercising outdoors and indoors in energy [270], fatigue [270], heart rate [81], and physiologic effects [271].

While considerable evidence suggests that participating in outdoor physical activity is good for mental health, there is some conflicting evidence. Klaperski et al. reported no difference in stress and wellbeing between indoor and outdoor exercisers [272]. Within the context of the COVID-19 pandemic, Colley et al. reported that individuals (n = 4524) who exercised outdoors reported better mental and physical health [273], while Folayan et al. reported that individuals (n = 4471) who exercised outdoors had greater odds of feeling lonely [274] and Jenkins et al. (n = 759) reported no significant relationship between exercising outdoors and mental health [275]. These studies were conducted in different countries (Canada, Nigeria, and Aotearoa), where public health measures and concerns about contracting the COVID-19 virus may have impacted many factors relating to physical activity including availability of locations to engage in exercise.

Intentionally interacting with nature may also benefit sleep. Multiple cross-sectional studies have correlated interactions with nature with greater sleep quality [117,207,255] and sleep patterns [118]. Time spent in nature and greater frequency of hiking has been positively associated with time asleep [178,251]. The effect that nature has on sleep can be immediate, as individuals who took a short walk in nature on their lunch break had greater restoration and sleep quality, as measured by heart rate variability, that night [276]. Vella et al. also reported improvements in sleep quality after their intervention [180].

Gardening is an activity that facilitates intentional interaction with nature. We acknowledge that there are multiple types of gardens including flower, water, and butterfly gardens, but for this review we only looked at fruit and vegetable gardening, which has been consistently associated with fruit and vegetable consumption [25,113,217,277,278,279]. Gardening has also been associated with greater frequency of consumption of fruits and vegetables [280]. When compared to their counterparts, individuals who garden more than four times per month report consumption of more fruits and vegetables [281]. This relationship is also reported in college students who garden more than once a week [282]. Among older adults, gardeners have greater odds of consuming fruits and vegetables five or more times per day [121]. Having access to a garden has also been associated with greater dietary diversity [283] and decreased frequency of food cravings and strength of the cravings, with these relationships being mediated by negative affect [35]. 

Gardening interventions have been successful in promoting fruit and vegetable consumption, in both quantity [284,285,286] and frequency [287]. For example, in a yearlong intervention among cancer survivors, 60% of the intervention group increased their fruit and vegetable consumption by more than or equal to one cup per day [288]. Blair et al. reported that participants of a nine month gardening program increased their fruit and vegetable consumption by 1.2 cups per day [289]. However, Tharrey et al. did not observe a significant change in fruit and vegetable consumption after participants gardened for one growing season [290].

## 4. Discussion

This narrative review presents evidence of how interactions with nature (indirect, incidental, and intentional) may be associated with mental and physical health status and health behaviors. As evidenced by the literature identified in this review, indirect and intentional interactions with nature have consistently been associated with better mental health and physical exhibitions of stress. Furthermore, intentional interactions with nature have been associated with engagement in physical activity and fruit and vegetable consumption. The following section discusses gaps in the literature and provides recommendations for future research.

In this review, we present evidence that interacting with nature (indirect, incidental, and intentional) may be associated with multiple components of health including decreased depression, anxiety, and stress and increased physical activity, sleep, and fruit and vegetable consumption. Health behaviors do not have separate effects on health outcomes; the culmination of behaviors leads to health outcomes. Leaders in behavioral medicine discuss that interventions should utilize multiple behaviors to have a greater impact on health [10,291]. Here, we provide evidence that engaging in physical activity while interacting with nature enhances the benefits of exercise including positive mental health and quality of life. Promoting healthy behaviors by interacting with nature may further promote mental and physical health outcomes. Therefore, healthcare providers should encourage patients to interact with nature to manage and promote positive mental and physical health status and behaviors.

A scarcity of research in the past decade has explored the relationship between incidental interactions with nature and health status and behaviors. Only 4% (n = 11) of papers identified focused on incidental interactions with nature, highlighting the need for future research to study this topic. A reason for these gaps in the literature may be due to the difficulty in measuring incidental interactions with nature [21]. The presence and care of houseplants was the method most often used for measuring incidental interactions. Future studies should consider measuring the frequency of walking through natural environments or working outdoors as ways to measure incidental interactions with nature.

Twenty-nine studies (10%) utilized longitudinal methods, highlighting the need for future studies using a longitudinal study design. In comparison to cross-sectional research, longitudinal research can be utilized to suggest if a variable has an effect on an outcome and can describe how patterns may change over time [292]. In this case, longitudinal studies could suggest whether interacting with nature influences mental and physical health status and behaviors. While experimental studies have shown that interacting with nature is associated with improvements in mental and physical health, many of these interventions were over short periods of time (minutes to weeks) and utilized mental health therapy within their experiment. Furthermore, utilizing longitudinal methods can aid in understanding how interacting with nature changes over the life course and answer other questions such as whether seasonality affects interactions with nature and health outcomes and behaviors.

While the relationships between interactions with nature and health behaviors have begun to be explored, there is still more to learn, especially from indirect and incidental interactions and sleep and dietary patterns. We were only able to identify 17 studies that examined the relationship between indirect interactions with nature and health behaviors and no studies that addressed a relationship between incidental interactions and health behaviors. Most studies in this area examined the relationships between intentional interactions with nature and physical activity. Exercise, sleep, and nutrition have been referred to as three additional pillars of health because each of these health behaviors promotes and maintains good physical health. It has been reported that engaging in physical activity is important for developing and maintaining good health and preventing diseases [293]. Suboptimal sleep quality and duration has been shown to deteriorate mental and physical health status including mood, anxiety, depression, obesity, immunity, and the development of various diseases [294].

Understanding the relationship between interacting with nature and dietary behaviors is important because consuming a high-quality diet is associated with decreased risk of developing diseases such as type 2 diabetes mellitus, cardiovascular disease, and certain cancers [295,296,297]. The average Healthy Eating Index-2020 score, which measures adherence to the Dietary Guideline for Americans 2020–2025, among adults is only 57 out of 100 and among older adults is only 61 out of 100 [298]. As mentioned previously, gardening has consistently been associated with fruit and vegetable consumption [25,217,277]. While fruits and vegetables are important components to healthy dietary patterns, other components include greater consumption of whole grains, dairy, protein, seafood, and unsaturated fatty acids and lowering consumption of refined grains, sodium, added sugars, and saturated fats [299]. Understanding how other interactions with nature may affect dietary behaviors can help guide the development of nature-based interventions aimed at improving dietary intake.

Using virtual reality, as an indirect way to interact with nature, is a promising approach for decreasing physiologic exhibitions of stress and promoting health. This approach may be particularly important for people with barriers that keep them from intentionally interacting with nature such as someone who is temporarily limited to indoor activities or living far from nature. As an indirect way to interact with nature, virtual reality allows people to immerse themselves in nature without having to be outside. While promising, few studies have assessed the acceptability of experiencing nature via virtual reality. In one study, when participants were prompted with a list of ways to indirectly interact with nature, breast cancer patients indicated that virtual reality was the least commonly enjoyed; instead, preferring watching nature through a window, viewing pictures of nature, and listing to nature [123]. However, the participants of this study had a mean age of 63.1 years and there may be greater acceptability of using virtual reality among younger individuals [123]. More research should explore the acceptability of using virtual reality as a way to indirectly interact with nature.

The findings of this narrative review supports the stress reduction theory (SRT) [300] and frameworks proposed by Markevych et al. [301] and Hartig et al. [302]. SRT proposes that physiological exhibitions of stress decrease when in nature [300]. In this review, we report that indirect and intentional interactions with nature have been associated with decreased physiological exhibitions of stress including cortisol, sympathetic nervous system activity, heart rate, and blood pressure. Markevych et al. and Hartig et al. proposed similar frameworks as to how the environment protects and promotes health. Notably, both frameworks add upon SRT by identifying that the natural environment can promote physical activity, thereby promoting health [301,302]. We add to their work by proposing that interacting with nature may promote additional healthy behaviors, such as sleep and dietary intake, though more research is needed to establish these relationships.

To the best of our knowledge, we are the first to review the relationships among interacting with nature (indirect, incidental, and intentional), mental health, physical health, and health behaviors, within the past 10 years. Notably, we were able to describe how indirectly and incidentally interacting with nature may promote health outcomes and behaviors. The methodology of our literature search was rigorous as we used specific Booleans, had inclusion and exclusion criteria, and had inclusion verified by a second author. We acknowledge several limitations within this article. First, this is a narrative review and not a systematic review. While we have discussed relationships that are reported throughout the literature, we did not evaluate the strength of the articles which were included as a systematic review and meta-analysis would. In addition, we only included papers published in English, limiting the generalizability of these results. Software which would be able to translate technical and scientific documents from other languages to English is still formative and should be verified with manual translation [303]. Since we would not be able to comprehend literature in the original language or with the assurance of manual translation, we decided to exclude papers published in other languages. While we included outdoor exercise in this review, per Keniger et al.’s work, not all the studies included may have measured exercise in nature. Finally, we only looked at the quantitative results in articles. Future reviews should consider synthesizing the qualitative research on the perceived benefits of interacting with nature.

## 5. Conclusions

In this narrative review, we synthesized the evidence of the relationships between interactions with nature and mental and physical health and behaviors, identified gaps in the literature, and provided recommendations for future research. While current evidence suggests positive relationships between (1) indirect and intentional interactions with nature and mental and physical health; and (2) intentional interactions and physical activity and fruit and vegetable consumption, there is a dire need for additional high-quality randomized control trials and longitudinal studies to further investigate these relationships. Furthermore, more research is necessary to understand the relationships between all types of interactions with nature, sleep, and dietary behaviors.

## Figures and Tables

**Figure 1 ijerph-21-00329-f001:**
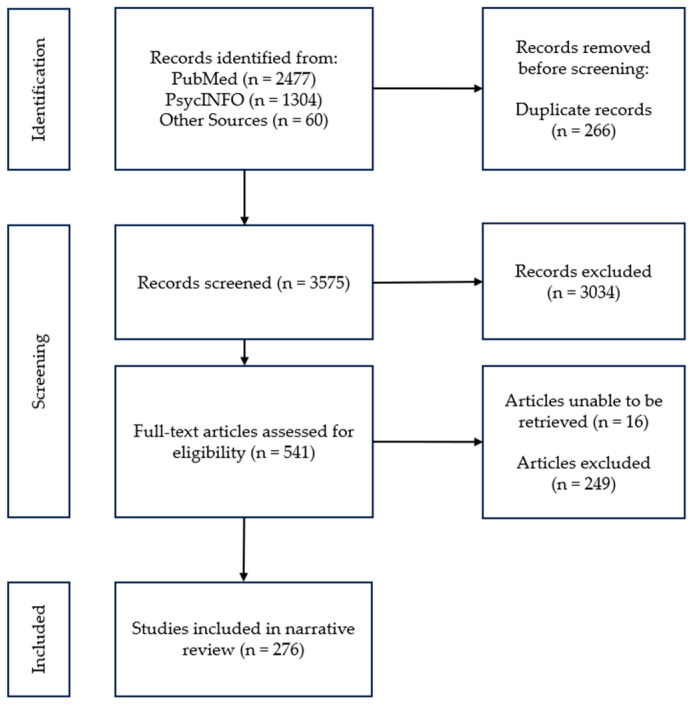
Flow diagram of article search and selection.

**Figure 2 ijerph-21-00329-f002:**
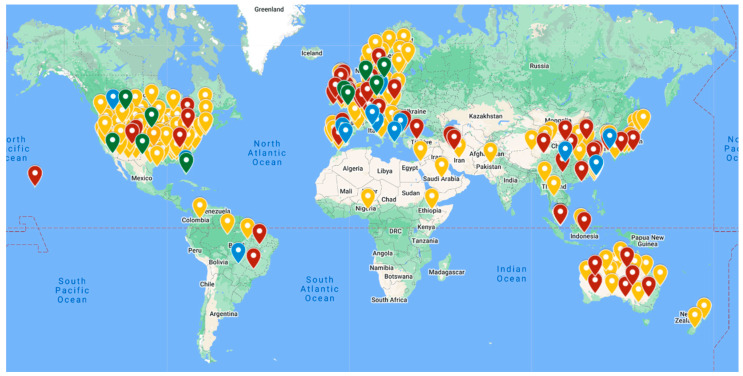
Types of interactions with nature studied across the world. Red pins represent indirect interactions with nature. Blue pins represent incidental interactions with nature. Yellow pins represent intentional interactions with nature. Green pins represent multiple types of interactions with nature studied.

**Table 1 ijerph-21-00329-t001:** Interactions with nature.

Type of Interaction with Nature	Definition	Examples and Select References	
Indirect	Experiencing nature without being in nature.	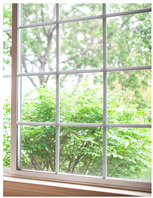	View of nature from indoors [13], pictures of nature [14], virtual reality [15], videos of nature [16], nature sounds [17], guided imagery [18], and perceived amount of greenspace [19].
Incidental	Experiencing nature, secondary to the main activity being performed.	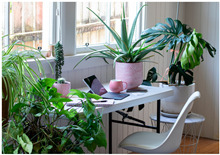	Indoor plants [20] and working outdoors [21].
Intentional	Purposeful interaction with nature.	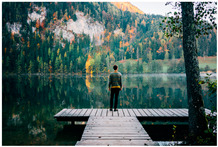	Spending time in nature [22], nature-based therapy [23], outdoor physical activity [24], and outdoor gardening [25].

## Data Availability

No new data were created or analyzed in this study. Data sharing is not applicable to this article.

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
