# Peer review of "Interactions with Nature, Good for the Mind and Body: A Narrative Review"

_ijerph, 2024, doi:10.3390/ijerph21030329_

Round 1

Reviewer 1 Report (Previous Reviewer 1)

Comments and Suggestions for Authors

This manuscript is greatly improved from the previous version (ijerph-2677605). The number of studies tabulated (placed in Supplementary Information, too massive, with 200+ studies, previously only 80+), as well as the discussion, are all comprehensively reviewed and summarized. Great job!

Author Response

REVIEWER 1

This manuscript is greatly improved from the previous version (ijerph-2677605). The number of studies tabulated (placed in Supplementary Information, too massive, with 200+ studies, previously only 80+), as well as the discussion, are all comprehensively reviewed and summarized. Great job!

Thank you for your comment! It is greatly appreciated!

Reviewer 2 Report (Previous Reviewer 3)

Comments and Suggestions for Authors

I would recommend this paper be moved forward to publication. I only found very minor items to consider reworking:

-Table 1 headers are not evenly spaced, it is a bit hard to tell what headers align with which material 

-For the search terms what is "AB" houseplants and "TI" houseplants- unsure what the AB/TI stand for when doing search terms

Thank you for the time and attention to revamping this paper. Figure 2 is fantastic- really well done, and the results are much more in-depth and valuable to readers based on your additions.

Author Response

REVIEWER 2

I would recommend this paper be moved forward to publication. I only found very minor items to consider reworking:

-Table 1 headers are not evenly spaced, it is a bit hard to tell what headers align with which material

Thank you for pointing out this oversight. The headers in table 1 (page 2) are now aligned to the bottom right of the cell. We do want to note that in ‘Examples and Select References’, we are providing visual and written examples.

-For the search terms what is "AB" houseplants and "TI" houseplants- unsure what the AB/TI stand for when doing search terms

Thank you for this comment. AB stands for abstract and TI stands for title. We have added these terms after the first instance of the abbreviation in the methods (page 3, line 102 and 103).

Thank you for the time and attention to revamping this paper. Figure 2 is fantastic- really well done, and the results are much more in-depth and valuable to readers based on your additions.

Thank you!

Reviewer 3 Report (Previous Reviewer 4)

Comments and Suggestions for Authors

I think that the authors have done a good job at addressing comments and the manuscript has benefited from the edits. Each section is stronger now, and the inclusion of many more key references has certainly situated the manuscript more clearly in the existing evidence base. I have two very minor comments that remain which the Editor may deem minor enough to not need addressing. I will leave that up to them:

1.       A small comment really just to clarify the novel findings in this work, but in lines 80-83, the authors say it is the first review that examines the three types of interactions with nature and their associations with mental and physical outcomes, as well as health behaviours. I think the authors need to be more explicit as to how their paper differs from the Keniger et al. paper given that review also focus on the three types of interaction with nature and on different levels of outcomes (psychological, cognitive, physiological, social, spiritual and other tangible benefits). Given there is focus in this manuscript on the outcomes wider than this and quite specific health behaviours like fruit and veg consumption, I do think this difference needs explicitly stated in order to support the claim in those lines.

2.       The methods section is much clearer and much improved. I do still have reservations about the exclusion of papers not in English but do appreciate the challenges the authors faced here with technical documents. I do think this rationale that was clearly communicated to the reviewers should be included in the methods or in the limitations (even briefly to explain the decision), as this comment is likely something other readers will pick up on and not be privy to the rationale provided to me.

Author Response

REVIEWER 3

I think that the authors have done a good job at addressing comments and the manuscript has benefited from the edits. Each section is stronger now, and the inclusion of many more key references has certainly situated the manuscript more clearly in the existing evidence base. I have two very minor comments that remain which the Editor may deem minor enough to not need addressing. I will leave that up to them:

  1. A small comment really just to clarify the novel findings in this work, but in lines 80-83, the authors say it is the first review that examines the three types of interactions with nature and their associations with mental and physical outcomes, as well as health behaviours. I think the authors need to be more explicit as to how their paper differs from the Keniger et al. paper given that review also focus on the three types of interaction with nature and on different levels of outcomes (psychological, cognitive, physiological, social, spiritual and other tangible benefits). Given there is focus in this manuscript on the outcomes wider than this and quite specific health behaviours like fruit and veg consumption, I do think this difference needs explicitly stated in order to support the claim in those lines.

      Thank you for pointing this out. We meant that in the previous 10 years, this is the only review that focuses on the three types of interactions with nature and health outcomes and behaviors. We have edited our statement on lines 80-83 (page 2) to be reflective of this.

  1. The methods section is much clearer and much improved. I do still have reservations about the exclusion of papers not in English but do appreciate the challenges the authors faced here with technical documents. I do think this rationale that was clearly communicated to the reviewers should be included in the methods or in the limitations (even briefly to explain the decision), as this comment is likely something other readers will pick up on and not be privy to the rationale provided to me.

      Thank you. We have added our rational for excluding articles published in other languages in the limitations section (page 14, lines 619-624).

This manuscript is a resubmission of an earlier submission. The following is a list of the peer review reports and author responses from that submission.

Round 1

Reviewer 1 Report

Comments and Suggestions for Authors

Review of ijerph-2677605

This review is very nicely written to correlate both indirect, incidental, and intentional interactions between humans and nature, as well as to classify the related gaps and future studies.

  1. Table 2 is the core of this manuscript. Please add row numbers in order to illustrate the size/gravitas of this review, and also to prevent the readers lost count. Please add numbers from 1 to 86 in Table 2.
  2. It seems that this is a systematic literature review, as indicated by the comprehensive summary of research in Table 2. Please add PRISMA diagram (Preferred Reporting Items for Systematic Reviews and Meta-Analyses http://prisma-statement.org/prismastatement/flowdiagram.aspx) to describe how the review of the 86 papers is obtained, such as how many original references were found, how many were cut, how many were retained, and the related explanations.
  3. It will be very nice to have an illustration of the Table 2 in form of a revised Figure 1, that is a world map but with the location is indicated by pinpoints labeled with the related soon-to-be-added row numbers of Table 2. Different types of the interaction with nature are then labeled with different colors of the pinpoint (or different colors of the number). In this way, as a picture is worth a thousand words, then this revised Figure 1 will be worth of 3300 words (more or less, of the 15-pages long Table 2).
  4. Table 2: Maund et al 2019 --> Six-week wetlands intervention… --> not “6-week…”
  5. Table 2: Ho et al 2022 --> Ten-day --> not “…10-day…”
  6. Table 2: Sahlin et al 2015 --> Sixteen-week of… --> not “16 weeks of” with no hyphenation.
  7. Table 2: Djernis et al 2021 --> Five-day residential… --> not “5 day” without hyphenation and with no plural version.
  8. Table 2: Irvine et al 2020 --> Twelve-week intervation --> not “12-week”
  9. Table 2: Sudimac et al 2022: Please write the definition of fMRI.
  10. Add list of all abbreviations and their definition, between Conclusion and References.
  11. References: Please write the references consistently. Some of the journal names are written unabbreviated, some are not. For example: this article is submitted to International Journal of Environmental Research and Public Health. The reference is sometimes written as “International Journal of Environmental Research and Public Health”, but sometimes as “Int J Environ Res Publich Health”. Please revise and write consistently.  

Author Response

REVIEWER 1

This review is very nicely written to correlate both indirect, incidental, and intentional interactions between humans and nature, as well as to classify the related gaps and future studies.

  1. Table 2 is the core of this manuscript. Please add row numbers in order to illustrate the size/gravitas of this review, and also to prevent the readers lost count. Please add numbers from 1 to 86 in Table 2.

Thank you for this recommendation. We believe that this recommendation is immensely helpful, especially as we have added many more papers to this review. Please see the revised Table 2 as the Supplemental Table.

  1. It seems that this is a systematic literature review, as indicated by the comprehensive summary of research in Table 2. Please add PRISMA diagram (Preferred Reporting Items for Systematic Reviews and Meta-Analyses http://prisma-statement.org/prismastatement/flowdiagram.aspx) to describe how the review of the 86 papers is obtained, such as how many original references were found, how many were cut, how many were retained, and the related explanations.

Thank you for this comment. We have added a PRISMA diagram as Figure 2. In addition, please see our edits to the first paragraph of 3.1 Overview of Identified Studies (page 4, lines 137-144).

  1. It will be very nice to have an illustration of the Table 2 in form of a revised Figure 1, that is a world map but with the location is indicated by pinpoints labeled with the related soon-to-be-added row numbers of Table 2. Different types of the interaction with nature are then labeled with different colors of the pinpoint (or different colors of the number). In this way, as a picture is worth a thousand words, then this revised Figure 1 will be worth of 3300 words (more or less, of the 15-pages long Table 2).

Thank you for this recommendation. We were not able to add the row numbers of the Supplemental Table into the map (now Figure 2, page 5) but were able to add pins and color those pins by the type of interaction with nature studied.

  1. Table 2: Maund et al 2019 --> Six-week wetlands intervention… --> not “6-week…”

Thank you, change made.

  1. Table 2: Ho et al 2022 --> Ten-day --> not “…10-day…”

Thank you, change made.

  1. Table 2: Sahlin et al 2015 --> Sixteen-week of… --> not “16 weeks of” with no hyphenation.

Thank you, change made.

  1. Table 2: Djernis et al 2021 --> Five-day residential… --> not “5 day” without hyphenation and with no plural version.

Thank you, change made.

  1. Table 2: Irvine et al 2020 --> Twelve-week intervation --> not “12-week”

Thank you, change made.

  1. Table 2: Sudimac et al 2022: Please write the definition of fMRI.

Thank you, change made.

  1. Add list of all abbreviations and their definition, between Conclusion and References.

Thank you for pointing this out, we have written abbreviations in their first appearance, per standard because we did not find an abbreviation section in the MDPI template.

  1. References: Please write the references consistently. Some of the journal names are written unabbreviated, some are not. For example: this article is submitted to International Journal of Environmental Research and Public Health. The reference is sometimes written as “International Journal of Environmental Research and Public Health”, but sometimes as “Int J Environ Res Publich Health”. Please revise and write consistently.  

Thank you for catching this, we have edited the references and ensured that the journal abbreviations are consistent throughout the references (page 18, lines 817 and 819).

Reviewer 2 Report

Comments and Suggestions for Authors

very interesting study, but the conclusions must be cautious because the narrative approach is not relevant to evaluate the impact of these approaches and interactions with nature, a selection of controlled RCT studies would be more appropriate and necessary to support conclusions precise -

Author Response

REVIEWER 2

Very interesting study, but the conclusions must be cautious because the narrative approach is not relevant to evaluate the impact of these approaches and interactions with nature, a selection of controlled RCT studies would be more appropriate and necessary to support conclusions precise –

Thank you for this important recommendation we have revised statements throughout the manuscript per your feedback and highlighted the need for high quality RCT’s and longitudinal studies in the conclusion section (page 14, lines 626 - 631)

Reviewer 3 Report

Comments and Suggestions for Authors

This work adds to the body of literature about interactions with nature and does so in a meaningful way by letting interaction speak to a broad array of experiences. 

-Some of the sentence structure was repetitive and lengthy; I would consider shifting sentence structure and varying length to help readers follow your writing. 

-Table 1 could be improved by sharing a visual representation fo each type to make a more comprehensive figure and see the differences between interaction types. 

-The introduction is missing past literature describing broadly the importance of this topic, why we should care about the 3 types of interactions how the different types, through virtual reality, indoor plants, etc., have been found to have health benefits. I found the introduction to be lacking in overall comprehensiveness of the body of literature + possible gaps. 

-The materials and methods discuss 3 classifications for mental health, physical health and health behaviors- would love to see a similar table to table 1 to describe these different outcomes, I think readers would want specific examples like what is provided in table 1 for these classifications in the introduction as well, and then a discussion on past literature + why it is important. 

-I am interested in all the keyword searches that you used. For example virtual reality is not discussed as a keyword but listed in table 1. I would also like a visual that shows how many articles were found, how many were excluded, etc. like a flowchart in review articles often have. I am unsure if all suitable articles are in this paper due to the unclear keyword search aspect. 

-Covid pandemic - 12 studies were conducted. Did the authors share that this was a COVID-19 pandemic study? I would reword to make this a bit more clear on if this was a research team decision or the article stated about COVID-19. 

-Please list the articles in alphabetical order within each interaction type and also share what health classification each article falls into. Curious why some are missing an intervention description- would want to clarify that in the text above the table and give a description of each if possible. 

-The results are very clearly presented and specific. My only issue is with how health behaviors are not really defined at the beginning of the paper. No idea of what keyword searches relate to health behaviors, and then it is unclear in the results what health behaviors are, which is very broad except for the result examples. Would consider rewording health behaviors to be a more specific title to what was examined. 

-Discussion and conclusion were clear + a helpful summary of results + implications. 

Comments on the Quality of English Language

No comments, minor edits to increase readability through using a service like Grammarly may be helpful, especially varying sentence structure. Would also check that all citations are in APA; noticed at least 1 article in the table is not in APA. 

Author Response

REVIEWER 3

Thank you, Reviewer 3 for your comments and suggestions for this manuscript. Please see below each of your comments for our response.

This work adds to the body of literature about interactions with nature and does so in a meaningful way by letting interaction speak to a broad array of experiences. 

-Some of the sentence structure was repetitive and lengthy; I would consider shifting sentence structure and varying length to help readers follow your writing. 

Thank you, we have worked on taking out redundancy and improve sentence structure.

-Table 1 could be improved by sharing a visual representation fo each type to make a more comprehensive figure and see the differences between interaction types. 

Thank you very much for this recommendation. We have added stock images to Table 1 (pages 2 and 3).

-The introduction is missing past literature describing broadly the importance of this topic, why we should care about the 3 types of interactions how the different types, through virtual reality, indoor plants, etc., have been found to have health benefits. I found the introduction to be lacking in overall comprehensiveness of the body of literature + possible gaps. 

Thank you for your comment. We have added several aspects to the introduction including the benefits of viewing nature, a description of health outcomes and behaviors (in reference to the below comment), and what previous reviews have explored in relation to the relationships among interacting with nature and health (pages 2-3, lines 51-67 and 78-83)

The materials and methods discuss 3 classifications for mental health, physical health and health behaviors- would love to see a similar table to table 1 to describe these different outcomes, I think readers would want specific examples like what is provided in table 1 for these classifications in the introduction as well, and then a discussion on past literature + why it is important. 

We have added details and examples of what health outcomes and behaviors are in the introduction: lines 60-67. Additionally, in these lines we discuss how engaging in multiple health behaviors results in certain health outcomes. After much discussion, our team decided not to include an additional table as to not overwhelm the reader. (The supplementary table with the main data is extremely long.) However, if the reviewer still feels that a table displaying the definition and examples of health outcomes and behaviors to be essential, we would be happy to reconsider.

-I am interested in all the keyword searches that you used. For example virtual reality is not discussed as a keyword but listed in table 1. I would also like a visual that shows how many articles were found, how many were excluded, etc. like a flowchart in review articles often have. I am unsure if all suitable articles are in this paper due to the unclear keyword search aspect. 

We have revised our methods with our specific Booleans and limiter. In addition, we added Figure 1 (page 4) with a PRISMA diagram of how articles were retrieved and selected. 

-Covid pandemic - 12 studies were conducted. Did the authors share that this was a COVID-19 pandemic study? I would reword to make this a bit more clear on if this was a research team decision or the article stated about COVID-19. 

The authors shared that the studies were conducted during the COVID-19 pandemic. We removed the COVID-19 column from the table (now Supplementary Table) but within the results still reference that some studies were conducted during the COVID-19 pandemic as this information provides important context to the findings.

-Please list the articles in alphabetical order within each interaction type and also share what health classification each article falls into. Curious why some are missing an intervention description- would want to clarify that in the text above the table and give a description of each if possible. 

Thank you for this recommendation. When you look at the Supplemental Table now, there is a column which describes the type(s) of interactions with nature the paper falls in. The papers are alphabetized within each type and each study type. 

-The results are very clearly presented and specific. My only issue is with how health behaviors are not really defined at the beginning of the paper. No idea of what keyword searches relate to health behaviors, and then it is unclear in the results what health behaviors are, which is very broad except for the result examples. Would consider rewording health behaviors to be a more specific title to what was examined. 

We have added a paragraph in the introduction (lines 60 – 67) which gives examples of healthy and unhealthy health behaviors and how they contribute to health outcomes.

-Discussion and conclusion were clear + a helpful summary of results + implications. 

Thank you!

Comments on the Quality of English Language

No comments, minor edits to increase readability through using a service like Grammarly may be helpful, especially varying sentence structure. Would also check that all citations are in APA; noticed at least 1 article in the table is not in APA. 

Thank you very much. We have ensured that all citations are correct (within the manuscript and supplementary table) and have improved the sentence structure.

Reviewer 4 Report

Comments and Suggestions for Authors

This is an interesting paper, and given the increased awareness of nature and human health and the number of studies now being published in the field, it is a timely contribution. I think the focus above and beyond the health behaviour outcome of physical activity is of particular note. I have a number of comments that I think need addressed prior to future publication.

Introduction:

The word ‘nature’ is not defined. In the context of this paper, does this term mean greenspace, bluespace, brownspace, rural areas, urban greenspace such as landscaped gardens, indoor vegetation, derelict and unused land, etc? If nature is being used as an umbrella term to cover all the above, then what is ‘not nature’. This lack of definition work around what ‘nature’ means has been raised by other research such as Taylor and Hochuli (2017) in their 'Defining greenspace: multiple uses across multiple disciplines' paper, and I think work in this field could be a lot clearer at defining what we mean when we are saying we are reviewing ‘nature and health’.  

The introduction is very brief, but it is very heavy on highlighting the benefit of ‘green prescribing’ type programmes. To highlight the range of benefits of nature, I think there should be at least some consideration/mention of literature around the evidence for both indirect and incidental interactions too, particularly since the focus of the review is not just intentional interactions.

In regard to green prescribing type programmes, there could be consideration that the concept of ‘green prescriptions’ is seen internationally and not just in North America. New Zealand have been doing it since the 90s, the UK have many programmes, Japanese and South Korean clinicians have been recommending forest bathing for decades, and countries in Scandinavia, such as Finland, have nature ‘dose’ recommendations too.  

Line 55 should be rephrased as it doesn’t make sense: “there has yet to be a review on the relationships among the three between these types of interactions with nature, mental and physical health and health behaviours among adults.”

Method:

The methods were really short, and I couldn’t see additional files with additional info? Apologies if I missed this. Full search strings need provided, not just examples of terms. Even though it is not a systematic review, the methods must be clear enough to be replicable, at the moment this is not the case.

Looking at the contributions, did only one researcher do the search and assess for inclusion/exclusion? If so, how did you address bias in selecting studies? I think that this is particularly important as it was not a fully systematic search so potentially more open to bias.

Why were articles that were published in another language excluded? Given the ability to translate articles easily using free online software, this is problematic to me. What was the reason behind this exclusion? If there is not a clear reason, this type of decision must be criticised for maintaining inequalities around the inclusion of Western-centric research over and above all other research. In the context of this study, this is particularly problematic, given the influence and importance of nature in non-Western countries. Keen to hear the justification of this decision, because currently the limitation section acknowledges this limitation, but there is still no reason as to why.

How did the authors decide on which mental and physical health outcomes and behaviors to include/exclude? The potential outcomes and behaviours here are so vast, it is hard to see how the authors would have only identified 86 studies in a ten-year period. They do say that studies were only included if there was explicit focus on individuals’ perception of nature, so what was the criteria for this – self report measures or something else? Just a bit more information on inclusion/exclusion in such a vast field would be helpful.

Relatedly, what exactly were the differences between outcomes and behaviours? I would have imagined there was quite a lot of overlap between these terms at times.

Results:

Why was there interest in which studies were conducted during Covid? Was it to suggest some findings weren’t ‘typical’ or another reason?

It seems a bit odd that only houseplants were considered as incidental interactions (and one working outside study). For example, were there no studies around active travel that fit into incidental outcomes? People may be passing through nature without intentionally interacting with it and this wasn’t considered at all. Relatedly, could this not also link to increased/changed health behaviours such as increased walking?

In the ‘health behaviours’ section of intentional interactions, the authors discuss how outdoor physical activity results in various mental health outcomes (references 77-80) and physical health outcomes (81). However, these studies are not included in the previous ‘mental health outcomes’ or ‘physical health outcomes’ section and I am curious as to why that is?

Discussion:

There was no discussion around the overlap between mental health outcomes, physical health outcomes, and health behaviours. In particular, I think it is quite important to consider if specific programmes or types of nature promote positive outcomes AND positive changes in behaviour as this would support the power of these types of programmes for population public health going forward.

There was limited linking with key existing studies in the literature. I know that the review focus was on the last ten years, but it was strange not to at least mention the idea of indirect benefits has been suggested for decades, right back to Ulrich’s hospital work with patients healing faster when beside a window.

There was no discussion around how this work maps onto previous work on theoretical pathways between nature and human health. For example, the well cited papers Nature and health by Hartig and colleagues (2014) with four proposed pathways and Markevych and colleagues’ paper ‘Exploring pathways linking greenspace to health: theoretical and methodological guidance’ were notably missing, particularly within the discussion. Although these papers use different language rather than ‘intentional interaction’, there are clear similarities between the work and, given the influence of these papers and their findings in the field, I do think it would be beneficial to consider including them, or at least mentioning them, in this section.  

Regarding limitations, the authors specify they ‘made assessments about the research as a whole’ but what does this mean/how did they do this if not using appraisal checklists as per systematic review? Were any papers excluded due to these assessments? This should be discussed in methods.

Author Response

REVIEWER 4

Suggestions for Authors

This is an interesting paper, and given the increased awareness of nature and human health and the number of studies now being published in the field, it is a timely contribution. I think the focus above and beyond the health behaviour outcome of physical activity is of particular note. I have a number of comments that I think need addressed prior to future publication.

Introduction:

The word ‘nature’ is not defined. In the context of this paper, does this term mean greenspace, bluespace, brownspace, rural areas, urban greenspace such as landscaped gardens, indoor vegetation, derelict and unused land, etc? If nature is being used as an umbrella term to cover all the above, then what is ‘not nature’. This lack of definition work around what ‘nature’ means has been raised by other research such as Taylor and Hochuli (2017) in their 'Defining greenspace: multiple uses across multiple disciplines' paper, and I think work in this field could be a lot clearer at defining what we mean when we are saying we are reviewing ‘nature and health’.  

Thank you for pointing this out. I’ve become more familiar with the discussions of the differences in greenspace or green space definitions. Our definition of nature is intentionally quite broad in this review to account for the different types of interactions with nature, we have included it in the first line of the introduction (page 1, line 32). I have also added clarify to the methods section and our inclusion and exclusion criteria (pages 3 - 4, Lines 112-123).

The introduction is very brief, but it is very heavy on highlighting the benefit of ‘green prescribing’ type programmes. To highlight the range of benefits of nature, I think there should be at least some consideration/mention of literature around the evidence for both indirect and incidental interactions too, particularly since the focus of the review is not just intentional interactions.

Thank you for pointing this out. We have added a paragraph which highlights Ulrich’s formative work on patient recovery when having a view of nature (page 2, lines 51-59).

In regard to green prescribing type programmes, there could be consideration that the concept of ‘green prescriptions’ is seen internationally and not just in North America. New Zealand have been doing it since the 90s, the UK have many programmes, Japanese and South Korean clinicians have been recommending forest bathing for decades, and countries in Scandinavia, such as Finland, have nature ‘dose’ recommendations too.  

Thank you for pointing this out. We have added some sentences that indicate interacting with nature is promoted across the world (pages 1-2, lines 43-48).

Line 55 should be rephrased as it doesn’t make sense: “there has yet to be a review on the relationships among the three between these types of interactions with nature, mental and physical health and health behaviours among adults.”

Thank you for pointing this out. We have revised that statement as “While there are existing reviews on the health benefits of exposure to nature, to the best of our knowledge, this is the first review that examines all three types of interactions with nature and their associations with mental and physical outcomes as well as health behaviors.” Found on page 3, lines 80 – 83.

Method:

The methods were really short, and I couldn’t see additional files with additional info? Apologies if I missed this. Full search strings need provided, not just examples of terms. Even though it is not a systematic review, the methods must be clear enough to be replicable, at the moment this is not the case.

We have added the Boolean strings and limiters to the Methods section (page 3, lines 92-111).

Looking at the contributions, did only one researcher do the search and assess for inclusion/exclusion? If so, how did you address bias in selecting studies? I think that this is particularly important as it was not a fully systematic search so potentially more open to bias.

We have decreased potential bias in selecting studies by reassessing the studies that were included and having the article inclusion checked by another author. Any disagreements to inclusion of a study were mediated by the senior author. This process is described on page 4, lines 124-130.

Why were articles that were published in another language excluded? Given the ability to translate articles easily using free online software, this is problematic to me. What was the reason behind this exclusion? If there is not a clear reason, this type of decision must be criticised for maintaining inequalities around the inclusion of Western-centric research over and above all other research. In the context of this study, this is particularly problematic, given the influence and importance of nature in non-Western countries. Keen to hear the justification of this decision, because currently the limitation section acknowledges this limitation, but there is still no reason as to why.

While some scoping reviews include works published in other languages, we were not able to identify any narrative reviews that do. While it is our eventual goal to be as inclusive as possible in a review, according to Nur Fitria (2021), the sophistication of translation software is still formative, especially for technical and scientific documents, and if utilized, should be checked with manual translation. In our opinion, it would be unethical to include literature in a narrative review that we cannot fully comprehend in the original or with the assurance of manual translation. Therefore, we regretfully acknowledge this limitation.

Nur Fitria, T. A Review of Machine Translation Tools: The Translation’s Ability. J Lang Lit 2021, 16, 162–176, doi:10.15294/lc.v16i1.30961.

How did the authors decide on which mental and physical health outcomes and behaviors to include/exclude? The potential outcomes and behaviours here are so vast, it is hard to see how the authors would have only identified 86 studies in a ten-year period. They do say that studies were only included if there was explicit focus on individuals’ perception of nature, so what was the criteria for this – self report measures or something else? Just a bit more information on inclusion/exclusion in such a vast field would be helpful.

Thank you for your comment. We did not exclude any health outcomes or behaviours, simply categorized them. While there are numerous measurements throughout this research, they can be categorized into mental health, physical health, and health behaviours. Yes, part of the inclusion included individuals perceptions of nature, a measurement on indirect interactions with nature. You can find an exhaustive list of inclusion and exclusion criteria on pages 4-5, lines 112 - 123. 

Relatedly, what exactly were the differences between outcomes and behaviours? I would have imagined there was quite a lot of overlap between these terms at times.

Health outcomes are measurements such as health status, depression, anxiety, stress, etc. Health behaviors are actions such as physical activity, sleep, dietary choices, time being sedentary, and smoking. These behaviors effect health outcomes. See page 2, lines 60-67.

Results:

Why was there interest in which studies were conducted during Covid? Was it to suggest some findings weren’t ‘typical’ or another reason?

We noted papers conducted during COVID-19 for a variety of reasons. First, across the world, individuals experienced greater deleterious mental health such as depression, anxiety and stress. Despite this, researchers have measured that interacting with nature, especially indirectly, was associated with better mental health. In addition, some findings were not typical such as Young et al and Jenkins et al, which may be attributed to the circumstances of the COVID-19 pandemic.

It seems a bit odd that only houseplants were considered as incidental interactions (and one working outside study). For example, were there no studies around active travel that fit into incidental outcomes? People may be passing through nature without intentionally interacting with it and this wasn’t considered at all. Relatedly, could this not also link to increased/changed health behaviours such as increased walking?

After redoing the methods, we were able to find one paper that measured active time traveling through nature. Most other papers focused on taking care of or having houseplants in the home. This was considered but not identified in the first submission of the paper. This certainly could be linked to increased walking, but we were not able to sparse whether authors of cross-sectional studies measured walking intentionally in nature or actively walking through nature, incidentally so we categorized those as intentional interactions with nature. For studies that had participants walk in nature as part of their intervention, we categorized those as intentional interactions with nature.

In the ‘health behaviours’ section of intentional interactions, the authors discuss how outdoor physical activity results in various mental health outcomes (references 77-80) and physical health outcomes (81). However, these studies are not included in the previous ‘mental health outcomes’ or ‘physical health outcomes’ section and I am curious as to why that is?

We kept these separate because we wanted to highlight that engaging in physical activity outdoors further promotes mental and physical health status, more than engaging in exercise indoors.

Discussion:

There was no discussion around the overlap between mental health outcomes, physical health outcomes, and health behaviours. In particular, I think it is quite important to consider if specific programmes or types of nature promote positive outcomes AND positive changes in behaviour as this would support the power of these types of programmes for population public health going forward.

Thank you for pointing this out. We have added to the discussion (page 12, lines 526-532) specifically mentioning that interacting with nature may promote health behaviors and outcomes.

There was limited linking with key existing studies in the literature. I know that the review focus was on the last ten years, but it was strange not to at least mention the idea of indirect benefits has been suggested for decades, right back to Ulrich’s hospital work with patients healing faster when beside a window.

Thank you for pointing out this oversight. We have decided to add this important formative work to the introduction instead of the discussion.

There was no discussion around how this work maps onto previous work on theoretical pathways between nature and human health. For example, the well cited papers Nature and health by Hartig and colleagues (2014) with four proposed pathways and Markevych and colleagues’ paper ‘Exploring pathways linking greenspace to health: theoretical and methodological guidance’ were notably missing, particularly within the discussion. Although these papers use different language rather than ‘intentional interaction’, there are clear similarities between the work and, given the influence of these papers and their findings in the field, I do think it would be beneficial to consider including them, or at least mentioning them, in this section.  

Thank you for these references. We have added a paragraph to the discussion comparing the results of this review to attention restoration theory, stress reduction theory, Markevych et al, and Hatig et al theoretical pathways. You can find this addition on page 13, lines 596 – 607.

Regarding limitations, the authors specify they ‘made assessments about the research as a whole’ but what does this mean/how did they do this if not using appraisal checklists as per systematic review? Were any papers excluded due to these assessments? This should be discussed in methods.

By using this phrasing, we meant that we highlighted relationships that are consistently reported throughout the literature. We have clarified this on page 14, lines 615-616. Because this is not a systematic review, we did not use appraisal checklists.